# Safety Monitoring System of CAVs Considering the Trade-Off between Sampling Interval and Data Reliability

**DOI:** 10.3390/s22103611

**Published:** 2022-05-10

**Authors:** Sehyun Tak, Seongjin Choi

**Affiliations:** 1Center for Connected and Automated Driving Research, Korea Transport Institute, 370 Sicheong-daero, Sejong 30147, Korea; sehyun.tak@koti.re.kr; 2Department of Civil Engineering, McGill University, 817 Sherbrooke Street West, Montreal, QC H3A 0C3, Canada

**Keywords:** safety monitoring system, cooperative intelligent transportation system, connected and automated vehicles, data reliability

## Abstract

The safety of urban transportation systems is considered a public health issue worldwide, and many researchers have contributed to improving it. Connected automated vehicles (CAVs) and cooperative intelligent transportation systems (C-ITSs) are considered solutions to ensure the safety of urban transportation systems using various sensors and communication devices. However, realizing a data flow framework, including data collection, data transmission, and data processing, in South Korea is challenging, as CAVs produce a massive amount of data every minute, which cannot be transmitted via existing communication networks. Thus, raw data must be sampled and transmitted to the server for further processing. The data acquired must be highly accurate to ensure the safety of the different agents in C-ITS. On the other hand, raw data must be reduced through sampling to ensure transmission using existing communication systems. Thus, in this study, C-ITS architecture and data flow are designed, including messages and protocols for the safety monitoring system of CAVs, and the optimal sampling interval determined for data transmission while considering the trade-off between communication efficiency and accuracy of the safety performance indicators. Three safety performance indicators were introduced: severe deceleration, lateral position variance, and inverse time to collision. A field test was conducted to collect data from various sensors installed in the CAV, determining the optimal sampling interval. In addition, the Kolmogorov–Smirnov test was conducted to ensure statistical consistency between the sampled and raw datasets. The effects of the sampling interval on message delay, data accuracy, and communication efficiency in terms of the data compression ratio were analyzed. Consequently, a sampling interval of 0.2 s is recommended for optimizing the system’s overall efficiency.

## 1. Introduction

The safety of urban transportation systems has been recognized as a public health issue worldwide; consequently, many researchers have contributed to improving safety performances. According to previous studies [1,2], approximately 90% of traffic incidents and accidents are caused by human factors. Therefore, many previous studies have used sensors and communication technologies to reduce the influence of human factors on traffic safety [3]. However, although researchers have focused on technologies using in-vehicle sensors, there exists several limitations in perception capability, detection range, and price in particular cases. Moreover, recent studies have concluded that connected and automated vehicles (CAVs) and cooperative intelligent transportation systems (C-ITS) can be a solution to ensuring transportation safety by reducing the perception and reaction error related to human drivers [4]. CAVs and C-ITS are based on information and communication technologies (ICT), which collect massive amounts of data from various sensors, transmit data and messages through telecommunications, process the data and messages to monitor real-time safety performances, and apply control strategies to vehicles and traffic management systems [5].

In the era of CAVs and C-ITS, there have been several changes in the role played by the transportation management center (TMC). First, traffic management is shifting from infrastructure to in-vehicle device-based management that relies on mobile devices and in-vehicle communication units. In particular, the previous research suggests that providing personalized traffic management can improve both traffic efficiency and safety compared to the conventional traffic management system, which provides traffic information based on the average behavior of road users [6,7,8,9,10]. Second, communication protocols in traffic management are witnessing a change from wired and mobile communications to vehicle-to-everything (V2X) and machine-to-machine (M2M) communications to ensure accurate and low-latency safety services. In the case of variable speed limit (VSL) and ramp metering, which are representative examples of conventional traffic management services, traffic control is performed using predefined schedules or transmitting a control method calculated in the server to the variable message signs (VMS) or traffic signals on the road through wired or mobile communication. However, in-vehicle device-based traffic management, using V2X or M2M communication, is more efficient than the conventional communication method, because it can provide warning information to the vehicle with low latency [11]. Finally, the nature of computing is changing from on-premise to cloud and edge computing to facilitate efficient and flexible data handling of multisource unstructured big data. To provide in-vehicle device-based traffic management in real time, various computing tasks such as data acquisition, data preprocessing, and control decisions must be performed efficiently. Therefore, it would be more efficient to subdivide computing roles (data acquisition, data preprocessing, communication, and data analytics) into a cloud and edge than to collect and process all data in one server, as in on-premise computing.

Thus, it is vital to develop a novel framework design, including data collection in-vehicle devices, transmission from vehicles to cloud servers, and data processing in cloud platforms, to properly monitor the safety of urban transportation networks and apply proper traffic control. However, a problem when developing and deploying such a framework is that the automated driving system produces too massive data every minute. The size of the produced data varies from 1 to 3 GB per minute, and these data consist of data points collected by various sensors, such as Lidar, radar, vision, chassis, and GPS. According to a previous study on the communication speed in South Korea, the average data rate of LTE communication is 23.6 Mbps, which is equivalent to approximately 1.4 GB per minute [12]. In addition, the maximum data rate of WAVE communication is 27 Mbps, which is equivalent to approximately 1.6 GB per minute [13,14,15]. Thus, both existing communication systems cannot guarantee sufficient data rates to transmit data from connected automated vehicles in real time for safety applications.

In the near future, ensuring higher-speed communication for C-ITS may be possible due to the emergence of 5G and 6G communication systems. However, considering the current deployment stage, the data size must be reduced through sampling to ensure that the data can be transmitted through the existing communication system. On the other hand, it is necessary to acquire highly accurate data to ensure the safety of different agents in C-ITS. Previous studies have also studied such a trade-off relationship in traffic data [16,17,18]. Thus, both communication efficiency and data reliability in data collection must be considered for new real-time safety applications to properly monitor the safety performances of connected vehicles (CVs) and automated vehicles (AVs). 

From research papers to standards, extensive studies have been undertaken on frameworks and architectures for future transportation management centers. In particular, many studies have focused on C-ITS, which enables direct communication between vehicles, roadside infrastructures (also known as roadside units, RSUs), and TMCs. A high-level system architecture was proposed in [19], comprising an end-to-end structure including a traffic control center, roadside infrastructure, and vehicle. In [20], a cloud-based vehicle information system architecture was proposed for C-ITS application services in connected vehicle environments. The use of the cloud system in C-ITS was proposed to process and store a massive amount of data generated from various sensors in C-ITS. Different types of data can be collected and analyzed under it, such as traffic flow and link speed data [21], safety-related data [22], and emission data [23]. With the emergence of CVs and CAVs, every vehicle traveling in a traffic network can function as a moving data generator and possibly merge with C-ITS. Furthermore, in [24], the exchange of traffic data, traffic control data, and traffic management strategies between public authorities and the private sector were analyzed with possible levels of cooperation under Traffic Management 2.0 (TM2.0). The exchange of data collected by sensors under the control of C-ITS and data collected by sensors installed in CV and CAV is the key to improving the overall performance of the traffic network. Although research covered an extensive range of studies with different aspects of C-ITS, many research gaps still exist for real-world deployment. The previous studies focused on developing the conceptual design of C-ITS, and as a result, it is difficult to use them directly. Additionally, previous studies did not consider the massive amount of data CAVs will produce when they emerge in urban transportation systems. Therefore, it is necessary to fill the research gap to properly design and deploy C-ITS for CAVs on the currently running communication networks by considering the trade-off between communication efficiency and data reliability. 

This study primarily focused on the safety performance of an urban transportation network with CAV and C-ITS. Safety applications and safety-related decisions can be handled by edge devices installed in CAVs [25]. For example, an emergency electronic braking lights application, one of the most common C-ITS services, generates the safety message in edge devices installed in CAVs when a driver abruptly brakes hard. These kinds of edge device approaches are useful when the types of safety indicators are constant and fixed. As the safety aspects for monitoring are diversified and increased, monitoring the safety performance of the overall network and confronting the possible fallbacks of the automated driving systems of CAVs are the roles that traffic management centers undertake. Certain studies [22,26,27] have proposed a TMC-based safety monitoring system that includes data flow. The results show that the TMC-based safety monitoring system provides a good opportunity for improving safety performances, because it can effectively consider the temporal and spatial relationship of the entire road network. However, they only consider the one type of service, so issues arising from increasing safety aspects that need to be monitored, such as communication efficiency and data reliability, for multiple applications have not been studied. 

Moreover, although the trade-off between communication efficiency through data sampling and data reliability has been studied in various fields [28,29], it has not been attempted in CAV and C-ITS. For example, in [18], a simplified resource allocation problem was defined to assess the trade-off between communication efficiency and data reliability under different sampling rates over a discrete time communication channel. Furthermore, Ref. [16] analyzed the effects of other variables, such as the penetration rate and sampling frequency of mobile sensors, on traffic state estimations in a highway scenario.

Thus, the objectives of this study are (1) to develop and design a C-ITS architecture and data flow for the safety monitoring systems of CAVs and (2) to determine the optimal sampling interval for data transmission considering the trade-off between communication efficiency and data reliability of safety performance indicators. Primarily, this study focuses on the design and deployment of C-ITS for CAVs, considering the trade-off of data reliability and communication efficiency.

The remainder of this paper is presented as follows. In Section 2 is a novel framework for traffic management centers for the safety monitoring of connected automated vehicles. In Section 2.1, the system architecture of the proposed traffic management center is presented with a detailed explanation of the data flow for the safety monitoring system of CAVs. Section 2.2 presents the safety performance indices used in this study. Section 2.3 presents an efficient data transmission technology from an onboard unit to a cloud server. In Section 3, the data used in this study and the experimental design to determine the optimal sampling transmission interval are introduced. In Section 4, the results of applying various sampling intervals for the three safety performance indicators are discussed. Finally, in Section 5, the contributions of this study and future study plans are presented.

## 2. Methodology

Safety monitoring of vehicles is one of the main issues in the field of transportation for both rapid responses to dangerous situations and a causal analysis of the dangerous situation [30]. There were two major approaches for the safety monitoring of various types of vehicles in the previous studies. The first approach was analyzing driving data after the end of driving [7,31,32,33]. The merit of this approach is that it can derive a meaningful analysis by handling a large amount of data. However, this approach is not applicable for real-time applications, because data collection activities are conducted at least once a day. The second approach is detecting dangerous situations with real-time data and is widely used in the field of C-ITS [34,35,36]. In this approach, the data size is strictly limited, such as 1 KB or less, considering the communication efficiency and its standard. It has the advantage of responding quickly to dangerous situations, but on the contrary, it has a limitation that the type of safety indicator that can be monitored is extremely limited, such as severe deceleration and sharp curves, due to the small message size. In some studies, to improve the utilization of limited small messages related to safety, dangerous situations are judged locally, and the results are transmitted to the TMC when the critical events are detected [37,38,39]. Since this method is limited in the scalability and flexibility of the safety indicators, it is hard to consider more diverse safety aspects than the designed function in the initial system plan.

With the introduction of CAVs, the requirements for the main functions of TMC are changing. First, the TMC for CAVs requires a faster response and more accurate analysis with more data than human-driven vehicles and connected vehicles. Second, considering the initial stage of research on CAVs, it requires a framework that makes it easy to introduce various safety indicators to respond to future safety issues in the middle of the system operations. Thirds, for the safety monitoring of CAVs, there is a need for what data items to be transmitted and how often they are transmitted among the vast amounts of data from CAVs. 

With the requirements for a safety monitoring system for CAVs, this study designed data flow for a TMC based on data collected from CAVs for safety monitoring. The data flow ranges from the data generated in sensors in the CAVs to the calculation of safety performances in a TMC. In particular, a data sampling method for transmitting the data collected from the in-vehicle sensors to the TMC via the roadside unit (RSU) has been proposed. The details are described in the subsequent subsections.

### 2.1. System Architecture

This study devises safety monitoring for CAVs, as shown in Figure 1. The proposed monitoring system comprises four systems: CAD system, onboard unit (OBUs) for communication, roadside units (RSUs) for communication, and TMCs for CAVs. The CAD system controls the longitudinal and lateral movements of vehicles based on the information of the sensors installed in the vehicle. In addition, the data from the in-vehicle sensors, such as chassis, vision sensor, radar sensor, Lidar sensor, and GPS, are first collected through the middleware such as the Robot Operation System (ROS) according to previous studies [40,41,42]. Each sensor has a different data collection cycle of less than 0.1 s, and the collected data are sent to the OBU for communication to be sent to the TMC for CAV.

The OBU sends the data collected from the CAD to the RSU. Since the data collected from the CAD comprises heterogeneous time intervals of various sensors, the unification of the heterogeneous time intervals from multiple sensors is necessary before sending the data to the RSU. Furthermore, data reduction is required due to the large size of raw data from the in-vehicle sensors that have to be sent over communications with limited transmission capabilities. Therefore, the OBU performs the sampling to unify the data received from the CAD simultaneously. For example, if the time interval of the raw data is 0.01 s, the data are converted to 0.1 s units via capturing the raw data once every 10 points during the sampling work. Moreover, the data volume is reduced by approximately 10% in this process, although the accuracy (reliability) of the data is reduced. Subsequently, the sampled data are packaged into the communication message format (V2X message) to be sent to the RSU [43] and, eventually, sent to the TMC. Simultaneously, the communication state information of the RSU is generated and sent to the TMC. 

In this study, a uniform sampling interval is only considered to reduce the computing load in the OBU. Applying different sampling intervals for each data can be beneficial in terms of the data size of a single message. For example, data closely related to safety are sampled every second, and data with low safety importance are sampled every two seconds. However, when different sampling intervals are applied, the sampling intervals must be determined for each data element, leading to an increase in the computing load in the OBU. Furthermore, different sampling intervals for each data element could decrease the overall reliability of the data collection, because data with a relatively long sampling interval may be designated as an optional type rather than mandatory for transmission of a message. 

The TMC performs functions to facilitate the safety monitoring of CAVs based on the data received from the RSU, which are primarily composed of the following four steps. First, the V2X message from the RSU is converted to a format that enables a more accessible analysis in a server, such as JSON, and then stored in the message broker, which can easily store and share data in real time. Second, a level of safety is calculated for each element of CAD to monitor the safety of the CAV based on the data stored in the message broker. This study used safety indicators for CAVs, such as severe deceleration, lateral position variation, and inverse time to collision. The safety indicators used are explained in the next section. Finally, the level of safety calculated through the safety indicators is displayed on the bulletin board to provide related information to the TMC manager. The safety index for each vehicle/road section is calculated and used for vehicle maintenance, performance improvement, and the warning of dangerous zones. Finally, the monitoring result for each vehicle/road section is transmitted to the CAV for safe driving. The CAV reduces the speed according to the monitoring results by adjusting the maximum speed for each road section. For example, when safety issues are detected in the TMC, it recommends that the CAV reduces the maximum driving speed from the road speed limit (e.g., 100 km/h) to a safe speed (e.g., 70 km/h).

### 2.2. Indicators for Monitoring Automated Driving Vehicles

For the monitoring of the safety performances of CAVs, various types of safety indicators are proposed and used in the previous research, such as the distribution of TTC at the brakes onset, the number of selected traffic violations, and the number of instances where the vehicle takes unnecessary collision avoidance action [44]. The proposed TMC is designed to cover all kinds of safety indicators; this study uses three safety indicators for monitoring different aspects of safety: severe deceleration (SD), lateral position variation (LPV), and inverse time to collision (ITTC). The SD is designed to capture unsafe and uncomfortable longitudinal driving movements and is among the extensively used safety surrogate measures for indicating the safety performances of both drivers and CAVs [45,46,47]. The occurrence of SD is determined by the existence or absence of the following condition:(1)$$Acc_{long}^{sub}\left(t\right)\leq D_{severe},$$
where $Acc_{long}^{sub}\left(t\right)$ is the longitudinal acceleration of the subject vehicle at time t, $D_{severe}$ is the threshold value for SD, which is set to −2.94 $m/s^{2}$ in this study, and explanation of the variables is shown in Figure 2.

This index comprehensively indicates the decreased comfort of users in the vehicle, the occurrence of an accident, and delayed cognition behavior of the preceding vehicle when there is a rapid deceleration of the vehicle. This is more strictly applied to mass transit, such as buses, because the safety of standing passengers in buses can be significantly decreased by the SD. 

The LPV represents vehicle safety related to lateral movement and is designed to capture the unsafe and unstable steering control of the lane-keeping system of connected and automated vehicles. The equation for the LPV is as follows:(2)$$I_{LPV}=\min\left(\left|y_{Left}^{dist}-\frac{l_{width}^{sub}}{2}\right|,\;\;\left|y_{Right}^{dist}-\frac{l_{width}^{sub}}{2}\right|\right),$$
where $y_{Left}^{dist}$ is the distance from the vehicle center to the left lane in the vehicle-moving direction, $y_{Right}^{dist}$ is the distance from the vehicle center to the right lane in the vehicle-moving direction, $l_{width}^{sub}$ is the width of the subject vehicle, and explanation of the variables is shown in Figure 2.

The gone over event, wherein a CAV approaches a lane too closely or crosses over a lane during driving, is a hazardous situation and among the most widely used evaluation factors in field tests [48,49,50]. The use of a LPV has been suggested to analyze such dangerous events numerically. It is considered a representative key performance indicator for evaluating the driving safety of CAVs, because it can detect a dangerous situation before the occurrence of a touched line event or gone over event [4,51].

The LPV determines whether the subject vehicle drives excessively to the left or right from the centerline. This implies that the minimum distance from the left or right side to the left or right lane, and the closer to zero this distance is, the more dangerous it is. For example, let us assume that a vehicle with a 2-m width is driving 0.5 m toward the right on the road with a lane width of 3.4 m. In this situation, the $l_{width}^{sub}$ is 2 m, $y_{Left}^{dist}$ is $\frac{3.2\;m}{2}+0.5$, and $y_{Right}^{dist}$ is $\frac{3.2\;m}{2}-0.5$. Consequently, the obtained value is: $I_{LPV}=\min\left(\left|2.1-\frac{2}{2}\right|,\;\left|1.1-\frac{2}{2}\right|\right)=\min\left(1.11,0.11\right)=0.11$. In this example, the vehicle is in a dangerous situation with a residual space of 0.11 m in the right direction. Consequently, the CAV must execute longitudinal vehicle control to return to the centerline or closely observe neighboring vehicles approaching from a near distance.

The ITTC represents the collision risk of the subject vehicle considering the distance and relative velocity between the subject and the preceding vehicles [52,53,54,55]. This indicator is often used to estimate the danger of the driver and CAV. The driver is provided a warning message to prevent an accident. The CAV performs control such as acceleration and braking based on this value to maintain a safe distance. This study used the ITTC to evaluate the ability of CAVs to recognize and appropriately respond to objects ahead. The equation for the ITTC is as follows:(3)$$I_{TTC}\left(t\right)=\frac{v_{sub}\left(t\right)-v_{pre}\left(t\right)}{d\left(t\right)},$$
where $v_{sub}\left(t\right)$ is the speed of the subject vehicle at time t, $v_{pre}\left(t\right)$ is the speed of the preceding vehicle at time t, $d\left(t\right)$ is the distance between the rear of the preceding vehicle and the front of the subject vehicle, and explanation of the variables is shown in Figure 2.

An ITTC value closer to 0 indicates a safer situation, and as it attains larger positive values, a more dangerous situation is indicated. Previous studies that used ITTC as a measure usually set the threshold to indicate a hazardous situation as 0.49 [38,41]. However, in this study, the safety performance indicators of the CAVs are the main focus, and thus, a threshold the same as human drivers should not be used. Considering previous studies, this study determined that an ITTC value larger than 1.76 is a dangerous situation [56,57].

### 2.3. Sampling for Transmission of Automated Driving Data

As previously discussed, the amount of data produced by CAVs every minute is approximately 1–3 GB. Under the current deployment stage, transmitting such large data is not possible via the existing communication networks in Korea, including WAVE (1.6 GB/min) and LTE (1.4 GB/min). Thus, as both communication systems cannot guarantee sufficient data rates to transmit data from connected automated vehicles in real time for safety applications, sample data must be collected before transmission through the communication network. The data collection frequency varies depending on the sensors and target variables, ranging from 20 to 50 Hz. This implies that the sensor can generate approximately 20–50 data points every second. Thus, if one data point is sampled every second, theoretically, the size of the data can be reduced to one-fiftieth, which the current communication network can accommodate for CAVs. 

Adjusting the data sampling interval as explained above can result in several benefits in terms of data transmission and storage. Still, the data accuracy decreases (error), and thus, a delay in detecting severe events can occur. Moreover, it is possible that severe events may not have been detected in some instances. Figure 3 presents examples of these problems. Figure 3a shows the delay and error that occurred when a sampling interval of 10 s was applied to the data collected in a 20 Hz unit. This figure graphically shows the acceleration of the CAV. At this moment, the automated driving monitoring system must detect the SD situation. However, due to the occurrence of a delay, because the system detected the peak point of the SD slightly late, an error was produced that underestimated the severity of this event. Figure 3b shows the result when a sampling interval of 5 s was applied to the data collected in a 20 Hz unit. The automated driving monitoring system failed to detect SD events in this situation.

The sampling interval may be increased, which reduces the data volume and gives data communication and storage; however, as shown in Figure 3, an error and delay in severe event detection can occur, and severe events may not be detected in certain cases. In contrast, the data volume increases if the sampling interval is decreased, which imposes a significant burden on data communication and storage. Thus, to satisfy these two requirements, a process for determining the optimal sampling interval is required.

Even if the sampled data are sent and stored by adjusting the sampling interval, the process becomes meaningless if the distribution significantly changes compared to the raw data. Therefore, before determining the optimal sampling interval, this study first analyzed whether the distribution of the sampled data was similar to the distribution of the raw data. Figure 4 shows examples of data distribution when data are acquired by applying a sampling interval as a probability density function. The black curve in each figure depicts the distribution of the values in the raw dataset. In contrast, the remaining curves depict the distribution of the values in the sampled dataset with different sampling intervals. The differences between the colored curves and the black curve in Figure 4 indicate that the data distribution change by sampling data points from the raw data. Consequently, to ensure that this distribution change does not cause statistical distributional change, the Kolmogorov–Smirnov test (KS test) is used, one of the extensively used statistical tests.

There are numerous options when choosing the correct statistical test, such as the *t*-test, Mann–Whitney test, and KS test [58]. In this study, the KS test is used, which is nonparametric and does not make assumptions about, or is not affected by, the original distribution of the data [59]. The KS test quantifies the distance between the empirical cumulative distribution function of two given data. For two particular data points, the equation for the KS statistic is as follows:(4)$$D_{n,m}=\underset{x}{\sup}\left|F_{1,n}\left(x\right)-F_{2,m}\left(x\right)\right|,\;$$
where $F_{1,n}$ and $F_{2,m}$ are the empirical distribution functions of the first and second samples, respectively, and sup is the supremum function.

Figure 5 shows the passing rate when the KS tests were performed 100 times. This result verifies whether the data obtained when sampling each datum exhibit a statistically similar distribution to that of the raw data. In the case of LPV, even a long sampling interval generates a statistically similar data distribution, but acceleration (longitudinal) cannot, because the KS test passing rate drops after 10 s. Thus, various sampling intervals can be applied for LPV, but longitudinal acceleration can generate statistically different distributions when a sampling interval of 10 s is used. Furthermore, using a long sampling interval should be avoided, because the ITTC data indicate a low passing rate, even when a short sampling interval is used.

## 3. Data and Evaluation

To analyze the effect of sampling intervals on the reliability of indicators and communication efficiency, this study collected data from CAVs and adjusted the sampling intervals of the collected data. In the data collection stage, a vehicle with various sensors was driven on a real road to obtain data for various dangerous situations. Subsequently, the effects of sampling intervals on the reliability of safety indicators and communication efficiency were analyzed at the effect analysis stage. Furthermore, regarding the reliability of the safety indicator, the reduction in the safety performance according to an increase in the sampling interval relative to the raw data was derived. The details are described in the following sections. 

### 3.1. Evaluation Metrics

As the sampling interval increases, the performance is improved in terms of the communication efficiency, because the amount of collected/transmitted data is reduced. However, it causes a reduction in the safety performance, because the safety-related events cannot be determined, as discussed in Section 2.1. Consequently, the safety performance in terms of the accurate detection of safety-related events and the communication efficiency in terms of the size of the data transmitted over communication exhibits a trade-off relationship.

The evaluation process of this study consists of three steps: the analysis of sampled points, analysis of the delay and error of the sampled data, and evaluation of the overall objective. In the first step, the characteristics of each safety performance indicator are investigated by analyzing the differences between the raw and sampled data points. The second step analyzes the distributions of the error and delay of safety-related events. Finally, in the last step, the overall performance of each sampling time interval is evaluated based on the objective function, considering both the communication efficiency and reliability of the safety indicator. The communication efficiency is evaluated in terms of the compression ratio (how much the data size is reduced) of the data and the reliability in terms of the detection success rate (how many critical events are successfully detected). The objective function used in this study is as follows:(5)$$\left(w_{com}\cdot x_{comp\;rate}^{ST=k}+w_{acc}\cdot x_{sucrate}^{ST=k}\right),$$
where $w_{com}$ and $w_{acc}$ are weights for the communication and accuracy of the safety indicator, respectively, $x_{comp\;rate}^{ST=k}$ is the compression ratio of the sampled data with a sampling interval (k s) compared to the raw data, and $x_{success\;rate}^{ST=k}$ is the detection success rate of the sampled data with a sampling interval (k s) compared to the raw data.

### 3.2. Experimental Environment

Figure 6 shows the sensor configuration and dimensions of the CAVs used for data collection. The CAV has a length of 6.195 m, a width of 2.038 m, and a height of 2.665 m. Seven Lidar sensors, two radar sensors, one vision sensor, and one GPS were installed in the CAV. The vision sensor collects image data from the vehicle. It generates information related to the objects and the environment, including the type of objects in front, the lane type, and the distance to the lane. At the same time, the radar sensors are used to detect objects ahead and behind and generate information such as the distance and relative velocity of a relatively far away vehicle. Furthermore, Lidar sensors generate information regarding the environment, such as safety facilities in the surroundings, and information about objects such as people and vehicles. Finally, the GPS generates information for estimating the locations of vehicles and time synchronization. All information was collected, time-synchronized, and stored through the ROS.

The data collected in this study includes PVSD (Probe Vehicle Safety Data) and AVSM (Autonomous Vehicle Safety Message). PVSD contains the information on the status of CAVs, including their GPS location, speed, and route. AVSM includes data collected by various sensors such as Lidar, radar, vision, chassis, and GPS. There are 64 specific data fields in PVSD and 118 data fields in AVSM. Among all the data we achieved, Table 1 shows a description of the data used in this study. Three safety indicators were used in this study, as discussed in Section 2.2. First, the SD is determined using the condition in Equation (1), which requires longitudinal acceleration data from the CAV. These data were collected using the Chassis sensor. Second, the LPV is determined using the condition in Equation (2), which requires the position of the left lane and the position of the right lane. These data fields are collected by a vision camera sensor installed in front of the CAV. In addition, ‘leftLaneQuality’ and ‘rightLaneQuality’ were used to filter out noisy data. Finally, the ITTC is determined using the condition in Equation (3), which requires the relative distance and relative speed between the ego vehicle and the obstacle in front of the CAV. These data were collected by the radar sensor installed in front of the CAV, and ‘targetRangeAccel’ (the relative acceleration) and ‘targetStatus’ were used to filter out noisy data.

The sensor data, including chassis, vision, and radar, were collected based on the test drives of CAVs conducted in Sejong City. Eight test drives were conducted from August to November 2020. The total length of the test route was 15.64 km, and the total driving time was 21 h. One test drive was divided into normal and abnormal driving sessions. In the normal driving session, the CAV traveled one predefined route without causing any safety-related events. In contrast, in the abnormal driving session, the CAV traveled the predefined route six times while generating safety-related events such as SD, left and right tilt, and rapid acceleration. The predefined route is shown in yellow line in Figure 7, and a sample set of the collected data can be found at: https://github.com/benchoi93/CAVTestDriveData (accessed on 6 May 2022). The architecture of this study is developed based on the cloud platform, Microsoft Azure. and the communications are based on WAVE.

## 4. Results and Analysis

### 4.1. Severe Deceleration (Longitudinal)

Figure 8 shows an example of the sampled points with different sampling intervals from 0.1 to 10 s. In the figure, the solid gray line and double black line represent the raw and sampled data, respectively. Furthermore, the black squares represent the points where the data were collected in the sampled data, indicated by the double black lines. In the example case in Figure 8, the vehicle initially drove at a constant speed with an acceleration near 0, rapidly decelerated at up to –3.4 m/s^2^, and returned to the acceleration of 0 m/s^2^. In this case, the amount of deceleration was increased for approximately 2 s, and subsequently, the amount of deceleration decreased for 10 s. The peak points occurred at 282.05 and 282.33 s. Moreover, as shown in Figure 8, as the sampling interval increased, the peak point of deceleration in the extracted data increased, which was delayed compared with the raw data. The peak points of the raw data occurred for the size of −3.4 m/s^2^ at 282.05 and 282.33 s. However, as the sampling intervals increased to 0.1, 0.2, 0.5, 1, 2, 5, and 10 s, the peak points tended to increase to −3.3, −3.3, −3.3, −3.2, −2.8, −2.7, −2, and −0.9 m/s^2^, respectively. Furthermore, the times when the peak points occurred were delayed at 282.47, 282.51, 282.49, 283.25, 283.19, 284.69, and 289.85 s. Overall, as the sampling interval increased, the data smoothened, and the detection was delayed.

Figure 9 shows the changes in the distributions of the delay and error at different sampling intervals. The figure shows the cases of detecting the occurrences of SD events (Figure 9a,c) and the cases of missing the occurrences of SD events (Figure 9b,d). Furthermore, Figure 9a,b represent the time difference between the peak point of the raw data and the peak point of the sampled data. This time difference represents the delay of SD event detection. The increase of the sampling interval exhibits different effects on the distributions in the detected and missed case. 

Figure 9a shows the delay distribution in the detected case. When the sampling interval increases from 0.1 to 10 s, the mode of the delay increases from 0.088 to 0.172 s, while the standard deviation increases from 0.041 to 1.30 s. Thus, when the sampling interval increases, the increase in the standard deviation is more clearly observed than the increase in the mode interval of the delay. This is related to the duration of the SD event, which is approximately 2.1 s on average. Therefore, if the sampling interval is larger than the average event duration, it will likely miss the SD event.

Figure 9c,d show the change in the error distribution of the sampled data at different sampling intervals. The effect of the sampling interval on the data distribution appears to be different in the detected and missed cases. In the detected case, when the sampling interval increases from 0.1 to 10 s, the mode interval increases from 1.087 to 1.263 m⁄$s^{2}$, while the standard deviation changes from 0.133 to 0.342 m/$s^{2}$. The change in the standard deviation is more significant than the change in the mode, because the points have a more significant difference between the raw and sampled data than the threshold value moves from a detected case to a missed case. Therefore, the mode of the detected case is restricted by the peak point value of the raw data and threshold value. In contrast, in the missed case, as the sampling interval increases from 0.1 to 10 s, the mode increases from 0.329 to 3.040 m/$s^{2}$. In the same situation, the standard deviation changes from 0.312 to 0.978 m/$s^{2}$, respectively. The changes in both the mode interval and standard deviation are large in the missed case. In contrast to the detected case, in the missed case, the sampled data are smoothed toward 0, and the change of the sampled data is decreased when the sampling interval increases. Consequently, it becomes far from the mode and standard deviation of acceleration of the raw data. 

Figure 10 shows the reliability and communication efficiency changes when different sampling intervals are applied. In this figure, the reliability of the indicators (in terms of the detection success ratio) is marked by a solid black line and circles. When this value is closer to 1, it implies that all critical events are detected, whereas, when closer to 0, fewer critical events than the raw data are detected. The communication efficiency (in terms of the compression ratio) is marked by a dotted black line and black diamonds. This value is the ratio of the reduced amount of sampled data to the amount of raw data. The closer it is to 1, the greater the size of the data reduction and vice versa. Finally, the weighted average of the detection success and compression ratios is marked by a black dotted line and black triangles, and the weight was set to 0.5 each.

As shown in Figure 10, when the sampling interval increased from 0.1 to 5 s, the detection success ratio decreased. However, the degree of the decrease was different. When the sampling interval changed from 0 (raw data) to 1 s, the detection success ratio decreased by 0.454, but when it changed from 1 to 5 s, it decreased by 0.329. In contrast to the detection success ratio, the compression ratio continuously increased with the sampling interval. However, the increasing trend appeared differently, depending on the sampling interval. When the sampling interval changed from 0 (raw data) to 0.2 s, the compression ratio increased by 0.905, but when it changed from 0.2 to 5 s, it increased by 0.091. Moreover, because of the different trends of the detection success and compression ratios, a peak point of the weighted sum of the two values was observed at 0.2 s. In other words, before 0.2 s, the rapid increase in the compression ratio had a significant effect on the weighted sum, thus increasing the weighted sum. However, after 0.2 s, the effect of the detection success ratio was more significant, thus decreasing the weighted sum. Considering these results, 0.2 s was recommended as the sampling interval for the SD.

### 4.2. Lateral Position Variation (LPV)

Figure 11 shows an example of the sampling points with different sampling intervals ranging from 0.1 to 10 s. In this figure, the solid gray and double black lines indicate the raw and sampled data, respectively. In the sampled data, the data marked by black squares represent the data collection points. The example case in Figure 11 shows a vehicle driving close to the right with an LPV of approximately 0.765 m and, further, moving close to the right with an LPV of 0.190 m at approximately 95 s. The margin to the right is approximately 0.190 m, which is a dangerous situation for lateral control, leading to a “touched line event”. In contrast to the case of SD in Figure 8, the error and delay of the LPV show different patterns. In terms of error, the peak point at all time sampling intervals is observed near 0.190. The difference between the peak point of the raw data and that of the sampled data is very small. In contrast, the time at which a peak point occurred was delayed as the sampling interval increased; that is, the error experienced minimal change, but the delay increased to 0.100, 0.182, 0.420, 0.827, 1.619, 3.864, and 7.041 s when the sampling interval increased to 0.1, 0.2, 0.5, 1, 2, 5, and 10 s, respectively. 

The error and delay show different patterns due to the duration of the event when a critical LPV appears. In the case of SD in Figure 8, the duration is only 2.1 s, whereas the duration of a critical event of LPV is approximately 19.99 s. Therefore, in the case of LPV, even if the delay increases due to a long sampling interval, it does not affect the detection of a critical event itself.

Figure 12 shows the changes in the distributions of delay and error at different sampling intervals. When the sampling interval increases from 0.1 to 10 s, the mode interval of the delay of the detected case (Figure 12a) increases from 0.16 to 8.57 s, while the standard deviation also increases from 0.043 to 3.561 s. This result is different from the result of the delay distribution of SD of the detected case (Figure 9a), where only the standard deviation changed, while the mode underwent a minimal change. In the case of LPV, the mode and standard deviation increase as the sampling interval increases because of the long duration of a critical event. In the detected case of LPV, the negative effect caused by increasing the sampling interval is reflected in the distribution of the delay. In addition, the effect of such a long duration was also observed in the missed cases. When the sampling interval increases from 0.1 to 10 s, the mode interval of the delay distribution of the missed cases (Figure 12b) increases from 0.109 to 6.374 s, while the standard deviation also increases from 0.030 to 2.661 s. The increasing trends of both the mode interval and standard deviation can be clearly observed. 

Due to the effect of the relatively long duration of the event, the impact of the sampling interval on the LPV error was not significant, as shown in Figure 12c,d; even when the sampling interval increased from 0.1 to 10 s, the mode interval of the error distribution of detected cases changed in small increments from 0.001 to 0.016 m. In addition, the standard deviation also varied from 0.005 to 0.016 m, which was the smallest of increments compared to the other indicators. Moreover, a similar trend was also observed in the error distribution of the missed cases. This was because the duration of the peak points of the critical event of LPV was often longer than the sampling interval. Even if the sampling interval increased, all dangerous situations have already been detected.

Figure 13 shows the changes in the reliability and the LPV’s communication efficiency of the LPV when the sampling interval is increased. As evident, in contrast to the SD case (Figure 10), the effect of the sampling interval change on the detection success ratio of the LPV is relatively small. Thus, at a sampling interval of 0.1 s, the detection success ratio is 0.999, whereas, at 10 s, it is 0.951. In contrast, the compression ratio increased significantly from 0.857 at 0.1 s to 0.974 at 10 s. Therefore, the sampling interval in the LPV has a much more significant effect on the compression ratio than on the detection success ratio. 

Owing to the contrasting effects of the sampling interval on the compression and detection success ratios, the time when the peak point of the weighted sum appears was also different from that of the SD. The peak value of the SD appeared when the sampling interval was 0.2, whereas that of the LPV appeared at 5 s. Thus, in the case of the LPV, even if a relatively long sampling interval was applied, dangerous situations can be detected more effectively than in the SD.

### 4.3. Inverse Time to Collision (ITTC)

Figure 14 shows an example of the sampling points, with different sampling intervals ranging from 0.1 to 10 s. In the figure, the solid gray and double black lines represent the raw and sampled data, respectively. Furthermore, in the sampled data indicated by a double black line, the black squares represent the data collection points. The example case in Figure 14 shows a change in the ITTC. A critical event with an ITTC higher than 2.995$\;s^{-1}$ occurred near 1718.88 s, and another critical event with an ITTC of 0.987$\;s^{-1}$ occurred near 1731.38 s. Considering that an ITTC value higher than 1.76${\;s}^{-1}$ is regarded as a critical event [41,42], the example case includes an urgent situation where an accident can occur if a prompt response is not made. 

Compared with the SD and LPV cases described earlier, the sampling interval of the ITTC had a significant effect on the detection accuracy. As shown in the 0.1-s sampling interval of Figure 14a, even when it was slightly increased by 0.1 s from the average time interval of 0.0499 s for the raw data, the dangerous situation of an ITTC of 2.995$\;s^{-1}$, which occurred near 1718.88 s, was not detected. Furthermore, a slightly dangerous situation of an ITTC of 0.987$\;s^{-1}$ at 1731.38 s, which maintained a relatively long duration, was barely detectable at sampling intervals longer than 1 s. Furthermore, certain situations were not detected at the sampling intervals of 0.1–0.5 s. Consequently, in contrast to the occurrence of a peak point value of 2.995$\;s^{-1}$ at 1718.88 s in the raw data, when the sampling interval was 0.1, 0.2, 0.5, 1, 2, 5, and 10 s, the peak points occurred at 0.987, 0.987, 0.987, 0.151, 0.127, 0.111, and 0.124${\;s}^{-1}$, respectively. Moreover, no dangerous situation was detected when the sampling interval was more significant than 1 s. 

The ITTC reacted more sensitively to an increase in the sampling interval than the other indicators, because the duration of the critical event of the ITTC was the shortest among the indicators. In the example case, the duration of the ITTC was around 0.1 s. Therefore, it failed to capture the peak points even if the sampling intervals were slightly increased. Due to the short duration of the ITTC, significant changes in the ITTC values were not observed when the sampling interval was more significant than 1 s.

Figure 15 shows the changes in the delay and error distributions, with different sampling intervals for the detected and missed cases. As shown in Figure 15a, in the delay distribution of the detected cases, when the sampling interval increased from 0.1 to 10 s, the mode interval increased significantly from 0.050 to 4.138 s. The standard deviation also increased significantly from 0.049 to 3.220 s. Similar significant increases in the mode interval and standard deviation were also observed in the delay distribution of the missed cases (Figure 15b). Due to the significant increase in the standard deviation according to the increase in the sampling interval, when the sampling interval was larger than 5 s, the uniform distribution that was almost flat was obtained. The significant increase occurred because, when the sampling interval increased, the critical event in the raw data can be barely detected, rather than detecting it even at a later time. Significant increases in the mode interval and the standard deviation are also observed in Figure 15c,d, indicating the error distribution changes with different sampling intervals. 

Figure 16 shows the changes in reliability and communication efficiency when the sampling interval increased. The figure shows that the detection success ratio decreased sharply when the sampling interval increased. In particular, when the sampling interval in the raw data increased to 0.1, 0.2, and 0.5 s, the detection success ratio decreased significantly to 0.502, 0.606, and 0.746. Subsequently, the decrease became smaller, and when the sampling interval became 10 s, the change was close to zero. Furthermore, the compression ratio, which represents the communication efficiency, increased significantly by 0.905, until the sampling interval was 0.5 s. Then, the change converged to 1, similar to the pattern of the other indicators. Due to the opposite changes in the detection success and compression ratios, the weighted sum of the two values reached a peak near 0.2 s. However, in contrast to the SD results (Figure 10), the changes in the weighted sum with different sampling intervals showed minimal variations between 0.5 and 0.6. This was due to the opposite tendencies of the detection success and compression ratio, which changed at similar values. This suggests that the effect of the sampling interval on the overall performance of the indicator was practically insignificant.

## 5. Conclusions

In this study, a safety monitoring system was proposed for connected and automated vehicles, and the effects of different sampling intervals on reliability and communication efficiency based on three indicators indicating various aspects of safe driving were analyzed. The analysis results showed that the increase in the sampling interval had similar effects on the communication efficiency for all three indicators. Although the degree of change depends on the size of raw data required for indicator calculations, all three indicators showed more than 90% improvement in their communication efficiency when the sampling interval was more significant than 0.5 s. Regarding the accuracy of the indicators, different results were observed for the three indicators. In the case of the SD, when the sampling interval was 0.5 s, the success ratio was approximately 0.7, showing an intermediate effect on the accuracy among the three indicators. In the case of LPV, the accuracy was 0.996 when the sampling interval was 0.5 s, and the success ratio changed very little because of the long duration of the critical event. Finally, in the case of the ITTC, the success ratio changed the largest to 0.254 when the sampling interval was 0.5 s due to the short duration of the critical event. Due to the different change patterns of the success ratio of the three indicators, the optimal sampling interval for each indicator was derived as 0.2 s for the SD, 5 s for the LPV, and 0.2 s for the ITTC. 

For efficient and reliable monitoring of the safety indicators, different sampling intervals should be applied to each indicator to obtain the optimal results. However, as shown in the data flow for the safety monitoring of CAVs in Figure 1, the communication messages transmitted from the CAD to the RSU and the center are designed for the integrated utilization of the safety monitoring of CADs, rather than for each indicator. Thus, designing and applying messages with different sampling intervals to the CAD monitoring system would probably decrease the integration efficiency due to the increased number of messages. Therefore, it is more advantageous to uniformly apply a sampling interval of 0.2 s to apply indicators with different characteristics. Furthermore, in the case of indicators with a long duration of the critical event, such as LPV, the increase in gain obtained by increasing the sampling interval was not significant compared to the other two indicators. For example, when the sampling interval was changed from 5 s to 0.2 s, the decrease in the weighted sum of the LPV was small at 0.061, whereas those of the SD and ITTC increased significantly by 0.253 and 0.045, respectively. To optimize the system’s overall efficiency in such heterogeneous features of safety indicators, a sampling interval of 0.2 s is recommended. 

In this study, although the sampling interval for safety monitoring was researched based on three safety indicators with different characteristics, it was also observed that the sampling interval suggested of 0.2 s could be changed depending on the features of the indicators. For example, suppose there were many indicators with a long duration, such as the LPV in the overall safety monitoring system. In that case, the optimal sampling interval will be larger than the results of this study. Therefore, when communication messages for the safety monitoring of CADs are designed, the reduction in the efficiency and the characteristics of the indicator should be considered. In the future, the characteristics of more diverse safety indicators will be analyzed by expanding the results of this study for various applications, such as evaluation of the collision risk and control of the automated vehicle, and for other execution strategies such as execution time and calculation of the safety indicator of an in-vehicle device [60,61,62]. Furthermore, although this study assumed the same sampling interval and data transmission interval, the follow-up paper will also analyze the optimal method for data transmission from connected and automated vehicles when different sampling intervals and transmission intervals, depending on their urgency, are applied.

## Figures and Tables

**Figure 1 sensors-22-03611-f001:**
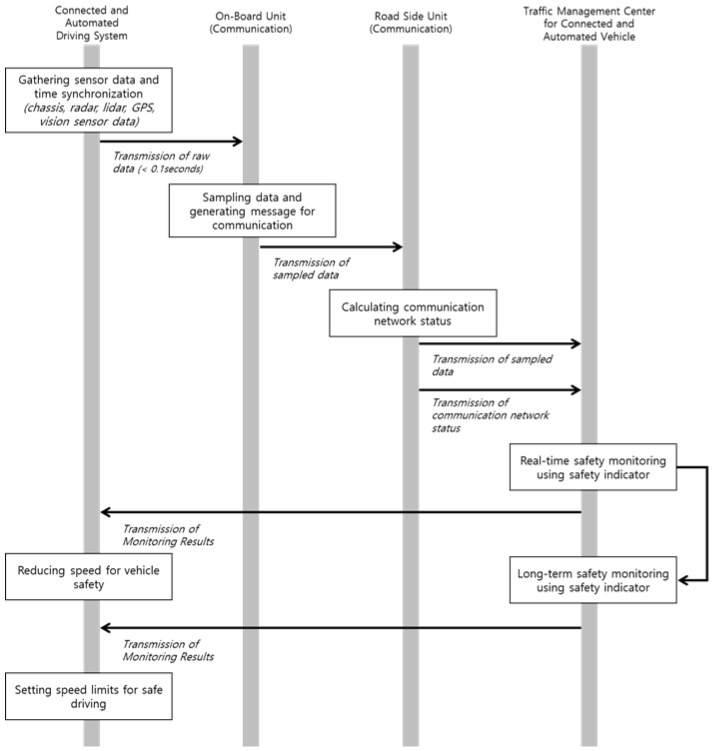
Data flow for the safety monitoring of connected and automated vehicles.

**Figure 2 sensors-22-03611-f002:**
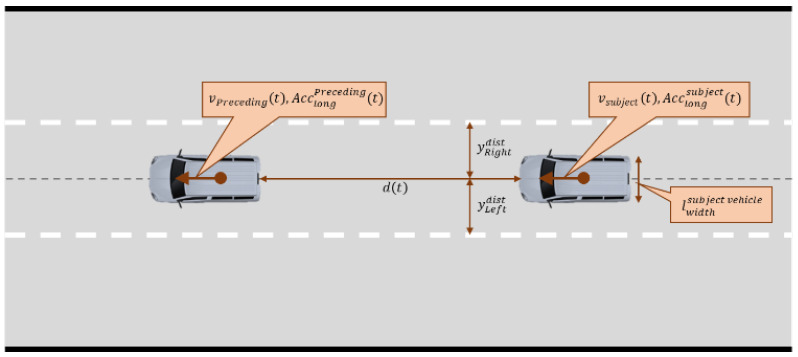
Explanation of the variables used to calculate the safety indicators.

**Figure 3 sensors-22-03611-f003:**
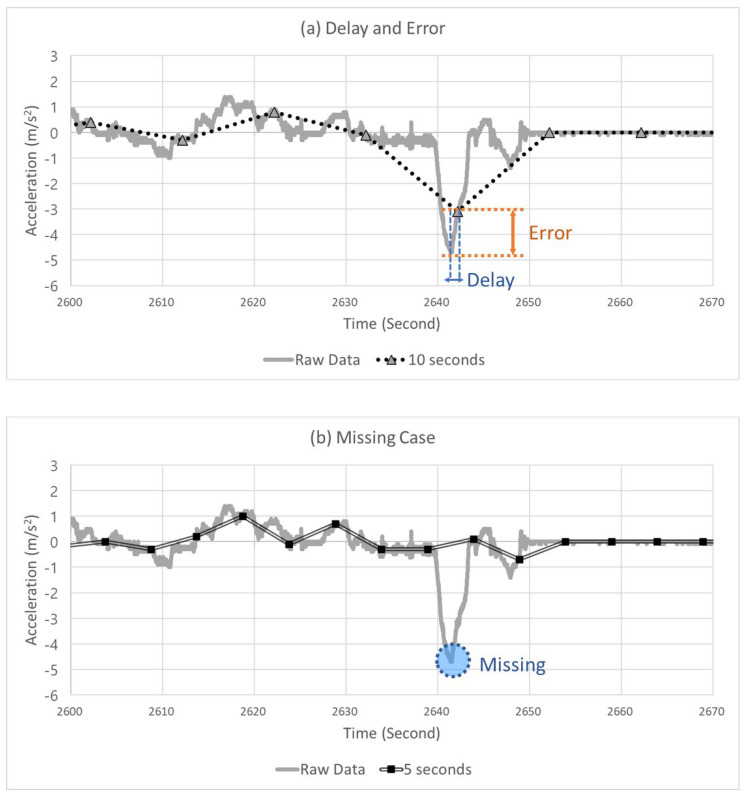
Example of (**a**) a delay and error case and (**b**) missing case.

**Figure 4 sensors-22-03611-f004:**
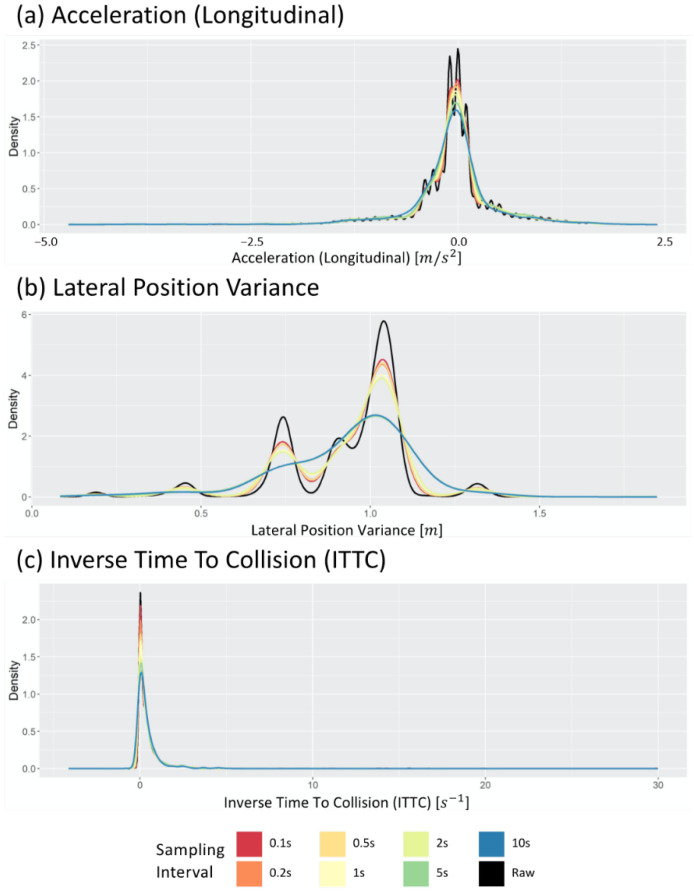
Change in distribution of (**a**) the longitudinal acceleration, (**b**) lateral position variation, and (**c**) inverse time to collision varying sampling intervals.

**Figure 5 sensors-22-03611-f005:**
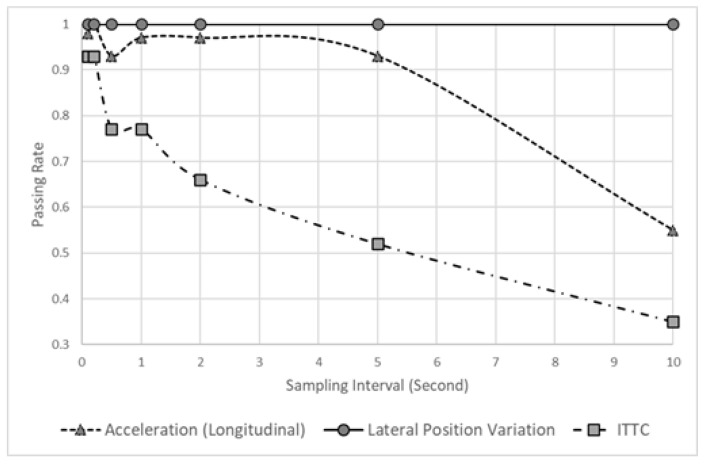
Passing rate of a hypothesis varying sampling interval.

**Figure 6 sensors-22-03611-f006:**
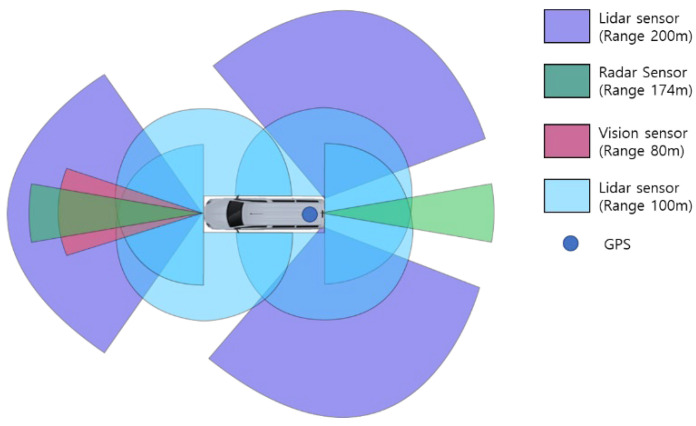
Sensor configuration of connected and automated vehicles for the experiment.

**Figure 7 sensors-22-03611-f007:**
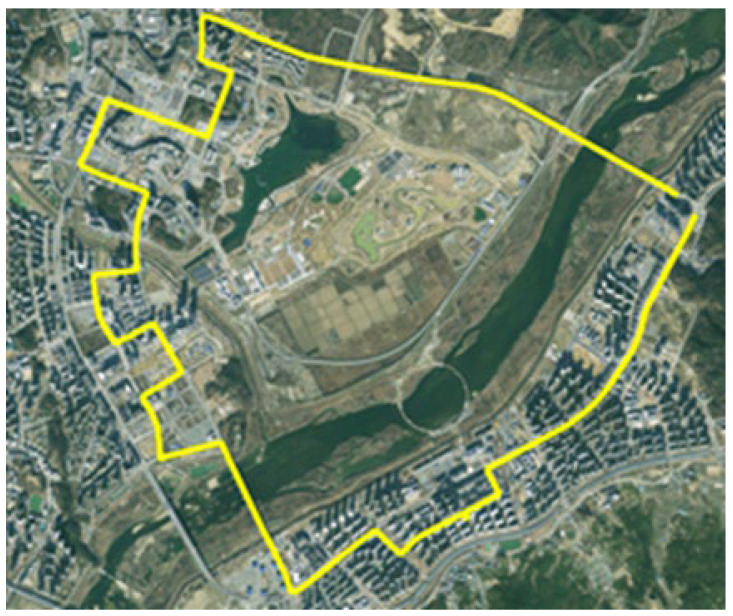
Predefined route for the test drives in Sejong City (Skyview source: Kakao map, National Geographic Information Institute).

**Figure 8 sensors-22-03611-f008:**
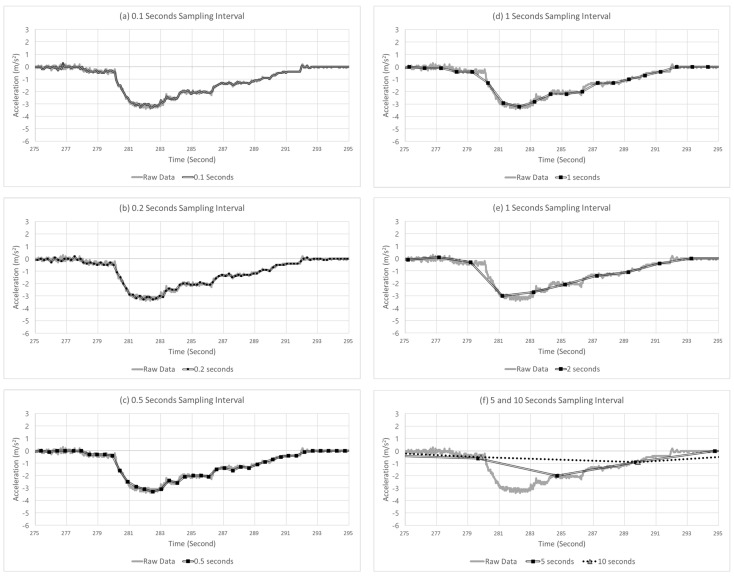
Example case of SD at varying sampling intervals.

**Figure 9 sensors-22-03611-f009:**
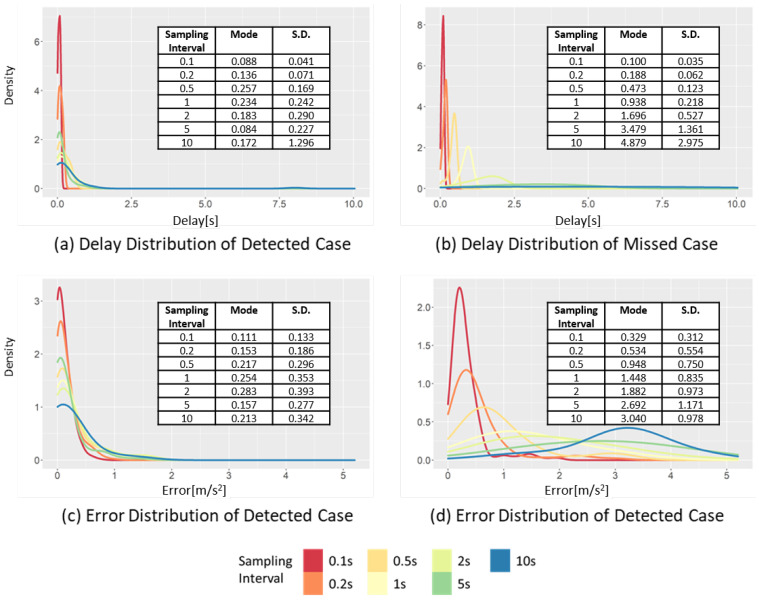
Changes in the delay and error distribution of the SD in the detected case and missed case, varying the sampling interval.

**Figure 10 sensors-22-03611-f010:**
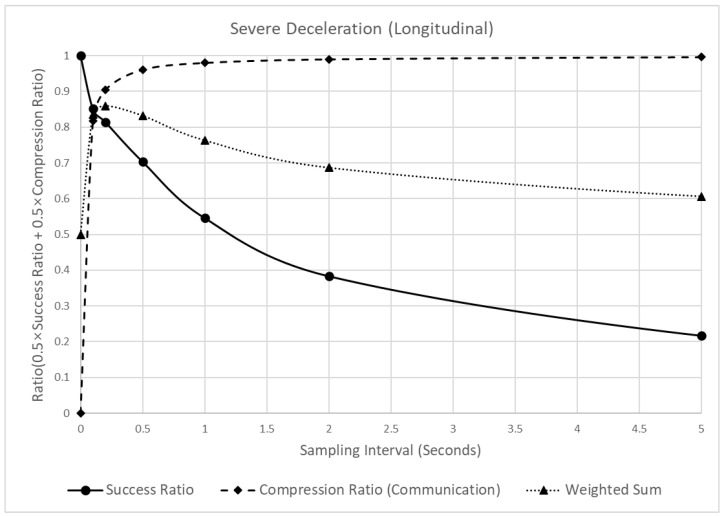
Analysis of the detection success ratio, compression ratio, and a weighted sum of the SD, varying the sampling interval.

**Figure 11 sensors-22-03611-f011:**
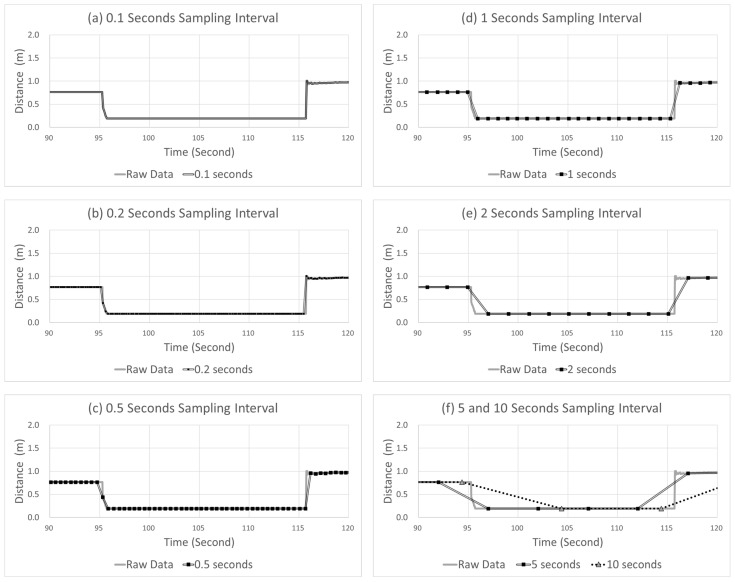
Example case of LPV varying sampling intervals.

**Figure 12 sensors-22-03611-f012:**
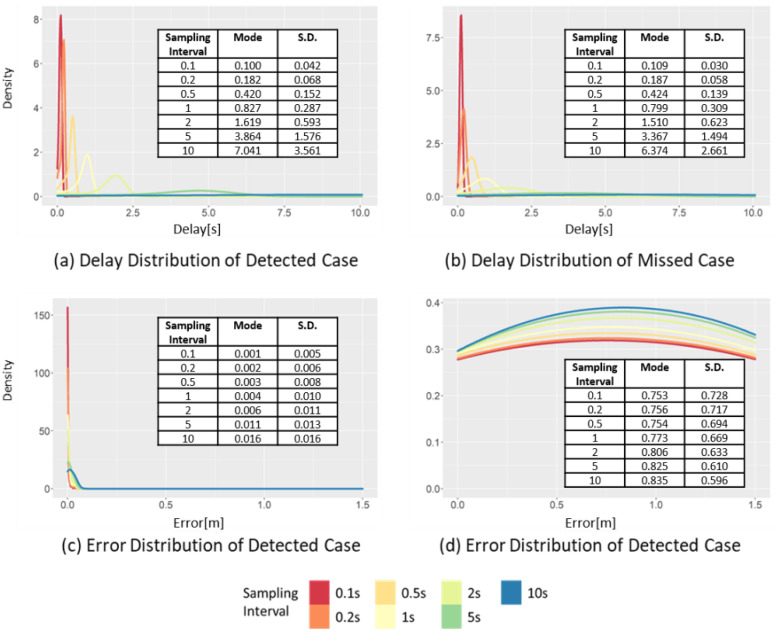
Changes in the delay and error distribution of the LPV in detected case and missed case, varying the sampling intervals.

**Figure 13 sensors-22-03611-f013:**
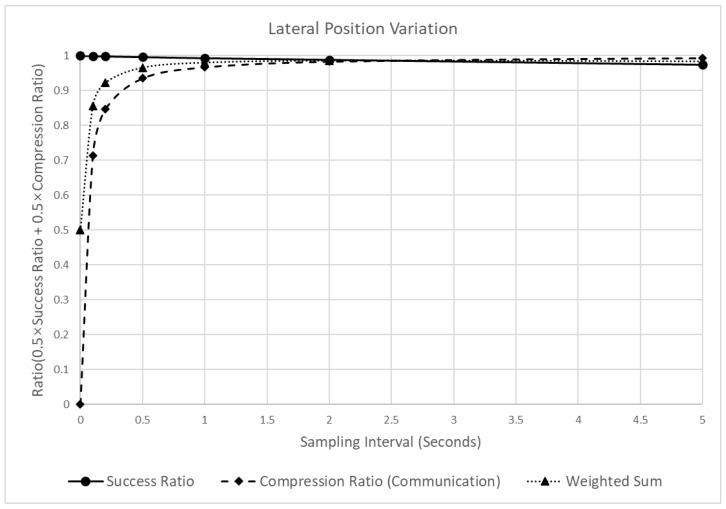
Analysis of the detection success ratio, compression ratio, and a weighted sum of the LPV varying the sampling intervals.

**Figure 14 sensors-22-03611-f014:**
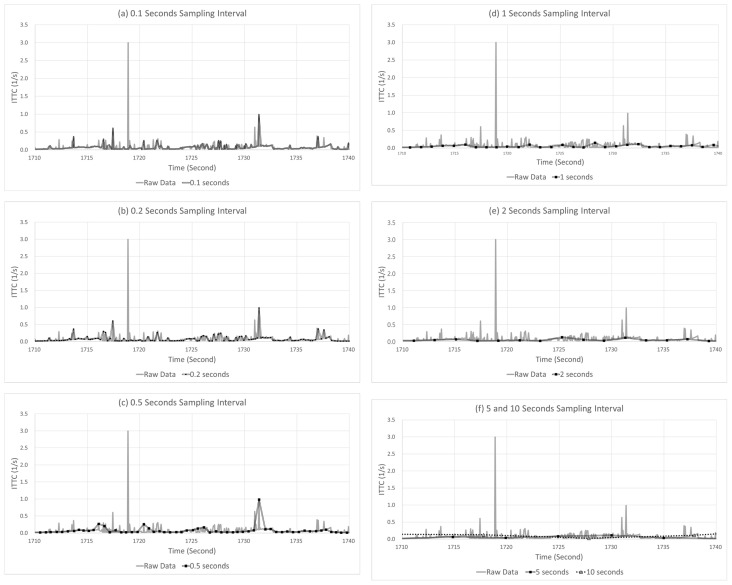
Example case of the ITTC varying the sampling intervals.

**Figure 15 sensors-22-03611-f015:**
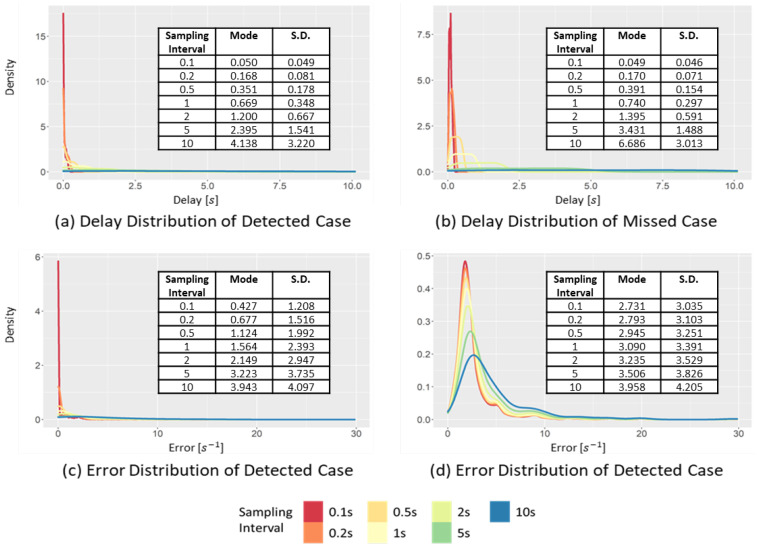
Changes in the delay and error distribution of the ITTC in the detected case and missed case, varying the sampling intervals.

**Figure 16 sensors-22-03611-f016:**
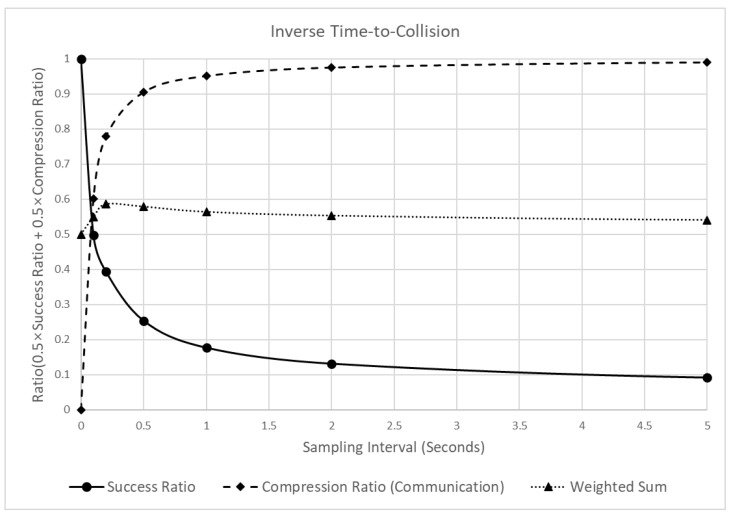
Analysis of the detection success ratio, compression ratio, and a weighted sum of the ITTC, varying the sampling intervals.

**Table 1 sensors-22-03611-t001:** Description of the data collected by the connected and automated vehicle.

Data Field		Description
timestamp		Data collection time (time synchronized through ROS)
Chassis	longAccel	Longitudinal acceleration of the ego vehicle
Vision	leftLanePosition	position of left lane from the ego vehicle
	leftLaneQuality	quality of left lane detection
	rightLanePosition	Position of the right lane from the ego vehicle
	rightLaneQuality	Quality of right lane detection
Radar	targetRange	Distance between target and the ego vehicle
	targetRangeRate	Relative speed between target and the ego vehicle
	targetRangeAccel	Relative acceleration between target and the ego vehicle
	targetStatus	Status of the target

## Data Availability

A sample set of the collected data can be found at: https://github.com/benchoi93/CAVTestDriveData (accessed on 6 May 2022).

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
