# Peer review of "Safety Monitoring System of CAVs Considering the Trade-Off between Sampling Interval and Data Reliability"

_sensors, 2022, doi:10.3390/s22103611_

Round 1

Reviewer 1 Report

Dear Authors,

the article entitled: Safety Monitoring System of CAVs considering the Trade-off between Sampling Interval and Data Reliability presents a novel Cooperative Intelligent Transportation System (C-ITS) architecture and data flow, including messages and protocols for the safety monitoring system of Connected Automated Vehicles (CAVs), and determined the optimal sampling interval for data transmission while considering the trade-off between communication efficiency and accuracy of safety performance indicators. It is very important because CAVs and C-ITSs are considered as solutions to ensure safety of urban transportation systems using various sensors and communication devices.

All chapters (abstract, introduction, methodology, data and evaluation, results and analysis, as well as conclusion) are very well described and they do not raise any doubts. In terms of the literature review is very sufficient (47 positions), all of which are papers from recognized scientific journals, such as: IEEE Transactions on Intelligent Transportation Systems, Journal of Advanced Transportation, Transportation Research Part C: Emerging Technologies, and others. Moreover, I would like to point out that the papers cited are related to the subject of this article (safety monitoring system, C-ITS, CAV and data reliability). However, in the publication make the following changes:

  • I propose to extend the literature in the following sentence of the methodology: „In addition, the data from the invehicle sensors such as chassis, vision sensor, radar sensor, lidar sensor, and GPS are first collected through the middleware such as the Robot Operation System (ROS)”, such as for example:
  1. Droeschel, D.; Schwarz, M.; Behnke, S. Continuous Mapping and Localization for Autonomous Navigation in Rough Terrain Using a 3D Laser Scanner. Robot. Auton. Syst. 201788, 104–115.
  2. Huang, L.; Chen, S.; Zhang, J.; Cheng, B.; Liu, M. Real-time Motion Tracking for Indoor Moving Sphere Objects with a LiDAR Sensor. Sensors 201717, 1932.
  3. Specht, M.; Specht, C.; DÄ…browski, P.; Czaplewski, K.; Smolarek, L.; Lewicka, O. Road Tests of the Positioning Accuracy of INS/GNSS Systems Based on MEMS Technology for Navigating Railway Vehicles. Energies 202013, 4463.
  • Please write sentences impersonally.

To sum up, after taking into account the above amendments (minor revision), I suppose that this article is suitable for publication in the Sensors.

Author Response

Thank you for your suggestions. We extended our literature reviews with the suggested previous works. Also, we revised the paper with impersonal sentences.

Reviewer 2 Report

This paper aims to study the impact of sampling intervals on the reliability of data collection of CAVs for safety monitoring. The analyses were conducted with empirically collected data. The reviewer has several major comments.

  1. The significance of safety monitoring system. This study is based on the assumption that the Traffic management centre needs to monitor the safety performance of CAVs and the limited transmission rate of data from the sensors to TMC. However, what is the significance of this safety monitoring system of TMC with sampled data as no actions required from the TMC. The CAVs can calculate the safety performance indicators locally and directly send the indicators to the TMC.
  2. Even if the TMC has the necessity of monitoring the CAVs, there is no need for CAVs to transmit all the generated data to the TMC for the purposes of safety evaluation. Critical variables related to the safety performance should be identified and assessed rather than sampling all the data. In addition, what is the original data size used for calculating the three safety indicators.
  3. Having said all, the performance of the sampling method shows poor results for the safety monitoring, as the success ratio for ITTC (the most critical indicator for safety) is only 0.5 with a sampling interval of 0.1s. This result is not satisfactory for the safety monitoring of CAVs.

Author Response

  1. This paper aims to study the impact of sampling intervals on the reliability of data collection of CAVs for safety monitoring. The analyses were conducted with empirically collected data. The reviewer has several major comments.

    1. The significance of safety monitoring system. This study is based on the assumption that the Traffic management centre needs to monitor the safety performance of CAVs and the limited transmission rate of data from the sensors to TMC. However, what is the significance of this safety monitoring system of TMC with sampled data as no actions required from the TMC. The CAVs can calculate the safety performance indicators locally and directly send the indicators to the TMC.

    Response from the authors

    Thank you for your comment. We will specifically respond to this comments in three parts.

    First, we feel that the previous version of manuscript didn’t fully described the aim of this study. We revised the paper to emphasize why it is required and important to monitor the safety performance of CAVs in TMC.

    Line 165

    Safety monitoring of vehicles is one of the main issues in the field of transportation for both rapid response to dangerous situations and causal analysis of dangerous situation. [28]. In the previous studies, there were two major approaches for the safety monitoring of various types of vehicles. First approach is analyzing driving data after the end of driving [7,29,30,31]. Merit of this approach is that it can derive a meaningful analysis by handling the large amount of data. However, this approach is not applicable for re-al-time applications because data collection activities are conducted at least one a day. Second approach is detecting dangerous situations with real-time data and widely used in the field of C-ITS [32,33,34]. In this approach, the data size is strictly limited such as 1KB or less considering the communication efficiency and its standard. It has the advantage of being able to respond quickly to dangerous situations, but on the contrary, it has a limitation that the type of safety indicator that can be monitored is extremely limited such as severe deceleration and sharp curve due to the small message size. In some studies, in order to improve the utilization of limited small size messages related to safety, dangerous situations are judged locally and the results is transmitted to the TMC when the critical events are detected [35,36,37]. Sine this method is limited in the scalability and flexibility of the safety indicators, it is hard to considering more diverse safety aspects than designed function at the initial system plan.

    With the introduction of CAVs, requirements for main functions of TMC are changed. First, TMC for CAVs requires a faster response and more accurate analysis with more data compared to human driven-vehicles and connected vehicles. Second, considering the initial stage of researches on CAV, it requires the framework that makes it easy to introduce various safety indicators to responds to future safety issues in the middle of the system operation. Thirds, for the safety monitoring of CAVs, there is a need for what data items to be transmitted and how often they are transmitted among the vast amounts of data from CAVs.

    Second, the purposes of safety monitoring system include both aspects of detecting real-time hazardous situations around CAVs and continuous maintenance of overall traffic network as we discussed in the previous version of the manuscript. To emphasize these purposes, we revised the manuscript and Figure 1 to include the examples of application of the proposed safety monitoring system as follows:

    Line 248

    Finally, monitoring result for each vehicle/road section is transmitted to CAV for safe driving. The CAV reduces the speed according to the monitoring results by adjusting the maximum speed for each road section. For example, when the safe issues are detected in TMC, TMC recommend that CAV reduces maximum driving speed from the road speed limit (e.g. 100km/h) to safe speed (e.g. 70km/h).

    Finally, as reviewer pointed out, we first considered calculating each safety performance index using edge computing in CAV. However, considering the researches in C-ITS and CAV are in early stage, there were many changes in safety performance depending on the control logics in CAVs. As a result, we decided to collect the data in the server for future decision-makings rather than calculating it in CAV. In addition, when the number of performance indicators increases, this will lead to the increase of computation and communication burden in CAV. In this case using data transmission from CAV to TMC, and calculating safety performance indicators in TMC would be more efficient. We revised our manuscript to fully discuss this according to your comments as follows:

    Line 120

    This study primarily focused on the safety performance of an urban transportation network with CAV and C-ITS. Safety applications and safety-related decisions can be handled by edge devices installed in CAVs [25]. For example, emergency electronic braking lights application, which is one of the most common C-ITS service generates the safety message in edge devices installed in CAVs when a driver abruptly breaks hard. These kinds of edge devices-approaches are useful when the types of safety indicators are constant and fixed. As the safety aspects for monitoring are diversified and increased,

    1. Even if the TMC has the necessity of monitoring the CAVs, there is no need for CAVs to transmit all the generated data to the TMC for the purposes of safety evaluation. Critical variables related to the safety performance should be identified and assessed rather than sampling all the data. In addition, what is the original data size used for calculating the three safety indicators.

    Response from the authors

    We agree with your comment. It is not necessary to store all data in the CAV in the center. And CAVs can calculate and transmit the necessary data according to the pre-defined set of safety performance indicators. However, as the requirements of safety monitoring for CAV increases, the communication burden will increase as well. As the researches on C-ITS and CAV are undergoing, it is not possible to determine necessary safety performance indicators at this point. This study proposes a good direction of managing data flows in such situations that it is necessary to continuously monitor and store safety performances. We revised the manuscripts to further discuss this as follows:

    Line 434

    The data collected in this study includes PVSD (Probe Vehicle Safety Data), AVSM (Autonomous Vehicle Safety Message). PVSD contains the information on status of CAVs including its GPS location, speed, route. AVSM includes data collected by various sensors such as lidar, radar, vision, chassis, and GPS. There are 64 specific data fields in PVSD and 118 data fields in AVSM. Among all data we achieved, Table 1 shows a description of the data used in this study.collected by the connected and automated vehicles in this ex-periment. Three safety indicators were used in this study, as discussed in Section 2.2. SD is determined using the condition in Equation (1), which requires longitudinal accelera-tion data from the CAV. These data were collected using the Chassis sensor. The LPV is determined using the condition in Equation (2), which requires the position of the left lane and the position of the right lane. These data fields are collected by a vision camera sensor installed in front of the CAV. In addition, ‘leftLaneQuality’ and ‘rightLaneQuality’ were used to filter out noisy data. Finally, ITTC is determined using the condition in Equation (3), which requires the relative distance and relative speed between the ego vehicle and the obstacle in front of the CAV. These data were collected by the radar sensor installed in front of the CAV, and ‘targetRangeAccel’ (the relative acceleration) and ‘targetStatus’ were used to filter out noisy data.

    1. Having said all, the performance of the sampling method shows poor results for the safety monitoring, as the success ratio for ITTC (the most critical indicator for safety) is only 0.5 with a sampling interval of 0.1s. This result is not satisfactory for the safety monitoring of CAVs.

    Response from the authors

    We fully agree with your comment. Especially, in the case of ITTC, it is suitable to identify and respond to vehicle safety immediately. However, in this study, it was conducted to continuously monitor the safety performance of vehicles and implement control strategies such as speed deceleration suitable for the characteristics of each road section rather than immediate response. In addition, the purpose of this study is to see how indicators show different characteristics according to communication issues such as sampling interval. To clarify this purpose, the paper was revised as follows. In addition, in future studies, studies will be conducted to supplement the TTC and the index showing characteristics, and these contents were reflected in the paper.

    Line 228

    For the monitoring of the safety performance of CAV, various types of safety indica-tors are proposed and used in the previous researches such as distribution of TTC at brake onsets, number of selected traffic violations, and number of instances where the vehicle takes unnecessary collision avoidance action. The proposed TMC is designed to cover these all kinds of safety indicators, this study uses three safety indica-tors for monitoring different aspects of safety

    Line 738

    In the future, the characteristics of more diverse safety indicators will be analyzed by expanding the results of this study for various applications such as evaluation of collision risk and control of automated vehicle and for other execution strategies such as execution time and calculation of safety indicator at in-vehicle device. Furthermore, although this study assumed the same sampling interval and data transmission interval, the follow-up paper will also analyze the optimal method for data transmission from connected and automated vehicles when different sampling intervals and transmission intervals depends on its urgent are applied.

Reviewer 3 Report

The paper “Safety Monitoring System of CAVs considering the Trade-off between Sampling Interval and Data Reliability” deals with a very high topics regarding the autonomous vehicle and safety of urban transportation. The paper introduces architecture and protocols for the safety monitoring system and determines the optimal sampling interval for data transmission while considering the trade-off between accuracy of safety performance indicators and communication efficiency, in terms of data volume.

The authors use three safety indicators that are acquired from a test drive and conducted, for each indicator, an extensive study about the effect of different sampling intervals on reliability and communication efficiency.

The paper is based on an extensive and recent references.

But the paper is very similar to another paper of the same authors: ”Development of Safety Monitoring System of Connected and Automated Vehicles Considering the Trade-Off between Communication Efficiency and Data Reliability”, available online at https://arxiv.org/ftp/arxiv/papers/2109/2109.12253.pdf and https://www.researchgate.net/publication/354950172_Development_of_Safety_Monitoring_System_of_Connected_and_Automated_Vehicles_considering_the_Trade-off_between_Communication_Efficiency_and_Data_Reliability

Can the authors provide more information about what is new/different/improvement in this paper?

*****

System Architecture and fig.1 shows data flow from CAV to Traffic management center. There is no data flow in opposite direction, form traffic management center to vehicle?

Fig. 4c: if you consider reducing x axis limits, more details will be visible.

What is the length (Km) of the test route?

Fig. 10, 13 and 15; y axis label: you wrote Rato instead Ratio

Fig. 8, 11 and 14 shows examples of the raw and sampled data for the considered safety indicators.

The question is: why do you consider a uniform sampling? If these indicators are critical for safety, why do you non consider a non-uniform sampling, and sending the information as soon as the respective indicator vary with a significant quantity relative to some normal value? In this case on can avoid missing situation as in fig. 3b and 8f. According to table 1 the data are accompanied by a time stamp. For safety reason and fast response, may be the first and immediate action can be taken by the vehicle itself?

Author Response

Reviewer 3

The paper “Safety Monitoring System of CAVs considering the Trade-off between Sampling Interval and Data Reliability” deals with a very high topics regarding the autonomous vehicle and safety of urban transportation. The paper introduces architecture and protocols for the safety monitoring system and determines the optimal sampling interval for data transmission while considering the trade-off between accuracy of safety performance indicators and communication efficiency, in terms of data volume.

The authors use three safety indicators that are acquired from a test drive and conducted, for each indicator, an extensive study about the effect of different sampling intervals on reliability and communication efficiency.

The paper is based on an extensive and recent references.

But the paper is very similar to another paper of the same authors: ”Development of Safety Monitoring System of Connected and Automated Vehicles Considering the Trade-Off between Communication Efficiency and Data Reliability”, available online at https://arxiv.org/ftp/arxiv/papers/2109/2109.12253.pdf and https://www.researchgate.net/publication/354950172_Development_of_Safety_Monitoring_System_of_Connected_and_Automated_Vehicles_considering_the_Trade-off_between_Communication_Efficiency_and_Data_Reliability

Can the authors provide more information about what is new/different/improvement in this paper?

Response from the authors

This paper is similar to “Development of Safety Monitoring System of Connected and Automated Vehicles Considering the Trade-Off between Communication Efficiency and Data Reliability,” available online at https://arxiv.org/ftp/arxiv/papers/2109/2109.12253.pdf. However, this is a preprint we uploaded on Arxiv before the submission. We confirm that this paper has not been published in another journal nor another conference proceeding.

*****

System Architecture and fig.1 shows data flow from CAV to Traffic management center. There is no data flow in opposite direction, form traffic management center to vehicle?

Response from the authors

Thank you for your comment. We didn’t include the opposite data flow in the previous version because we thought it was beyond the scope of this study. We had data flows from TMC to CAVs too in our full framework. As a result, according to the reviewer’s comment, we included the opposite data flow in Figure 1 and revised the manuscripts as follows:

Line 248

Finally, monitoring result for each vehicle/road section is transmitted to CAV for safe driving. The CAV reduces the speed according to the monitoring results by adjusting the maximum speed for each road section. For example, when the safe issues are detected in TMC, TMC recommend that CAV reduces maximum driving speed from the road speed limit (e.g. 100km/h) to safe speed (e.g. 70km/h).

Line 183

With the introduction of CAVs, requirements for main functions of TMC are changed. First, TMC for CAVs requires a faster response and more accurate analysis with more data compared to human driven-vehicles and connected vehicles. Second, considering the ini-tial stage of researches on CAV, it requires the framework that makes it easy to introduce various safety indicators to responds to future safety issues in the middle of the system operation. Thirds, for the safety monitoring of CAVs, there is a need for what data items to be transmitted and how often they are transmitted among the vast amounts of data from CAVs.

Fig. 4c: if you consider reducing x axis limits, more details will be visible.

What is the length (Km) of the test route?

Response from the authors

The total length of test route is 15.64 km. We revised the manuscript as follows:

Line 452

The sensor data, including chassis, vision, and radar, were collected based on the test drives of CAVs conducted in Sejong city. We conducted eight Eight test drives are con-ducted from August to November 2020., The total length of test route is 15.64 km and the total driving time was 21 hours. One test drive was divided into two sessions: normal driving and abnormal driving. In the normal driving session, the CAV traveled one prede-fined route without causing any safety-related events. In contrast, in the abnormal driving session, the CAV traveled the pre-defined route six times, while generating safety-related events such as SD, left and right tilt, and rapid acceleration. The pre-defined route is shown in Figure 7, and a sample set of the collected data can be found at: https://github.com/benchoi93/CAVTestDriveData. The architecture of this study is devel-oped based on the cloud platform, Microsoft Azure. and the communications are based on WAVE.

Fig. 10, 13 and 15; y axis label: you wrote Rato instead Ratio

Response from the authors

Thank you for pointing out the typos. We revised the Figures.

Fig. 8, 11 and 14 shows examples of the raw and sampled data for the considered safety indicators.

The question is: why do you consider a uniform sampling? If these indicators are critical for safety, why do you non consider a non-uniform sampling, and sending the information as soon as the respective indicator vary with a significant quantity relative to some normal value? In this case on can avoid missing situation as in fig. 3b and 8f. According to table 1 the data are accompanied by a time stamp. For safety reason and fast response, may be the first and immediate action can be taken by the vehicle itself?

Response from the authors

Thank you for your insightful comment. Our previous version didn’t fully explain why we chose to use uniform sampling. There were many technical difficulties of using various sampling rates depending on the indicator type. Especially the computing load in the OBU was the major problem. To further describe this issue, we revised the paper as follows:

Line 228

In this study, uniform sampling interval is only considered to reduce the computing load in the OBU. In terms of data size of single message, applying different sampling intervals for each data can be beneficial. For example, data closely related to safety is sampled every second and data with low safety importance is sampled every two seconds. When different sampling intervals applied, the sampling cycle must be determined for each data element, and it leads to an increase in the computing load in the OBU. Further-more, different sampling interval for each data element could result in decrease in overall reliability of data collection because data with a relatively long sampling interval may be designated as an optional type rather than a mandatory for transmission in a message.

Line 738

In the future, the characteristics of more diverse safety indicators will be analyzed by expanding the results of this study for various applications such as evaluation of collision risk and control of automated vehicle and for other execution strategies such as execution time and calculation of safety indicator at in-vehicle device [58,59,60]. Furthermore, although this study assumed the same sampling interval and data transmission interval, the follow-up paper will also analyze the optimal method for data transmission from connected and automated vehicles when different sampling intervals and transmission intervals depending on its urgency are applied.

Also we further discussed this in the Conclusion

Reviewer 4 Report

This paper presents safety monitoring system of CAVs considering the trade-off  between sampling interval and data reliability.The investigated topic is interesting, the experimental results are used to illustrate the effectiveness of the proposed  algorithm.  Generally,  the  paper  is  well  written  and  organized.  Thus,  I  will  recommend  several  clarifications  before acceptance of this paper which are listed below: 1. According to the topic of this paper, the authors need to supplement some related references to emphasize the main contribution and innovation of the paper as for the effect of sampling interval and data transmission for active safety system of advanced control technique such as Simultaneous Feasible Local Planning and Path-Following Control for Autonomous Driving,Robust vibration control for active suspension system of in-wheel-motor-driven electric vehicle via μ-synthesis methodology. 2.The detailed test platform and its layout, the detailed process of the test should be presented.3.The expression should be improved. The English needs an improvement. The paper has some typos which need to be checked and corrected in the revision.4. The entire flowchart and presentation of results should be improved, these results are not clear. 5.The use of proposed sampling interval and data transmission algorithm rather than other techniques should be discussed,execution time comparison in tests should be provided.

Author Response

Reviewer 4

This paper presents safety monitoring system of CAVs considering the trade-off  between sampling interval and data reliability. The investigated topic is interesting, the experimental results are used to illustrate the effectiveness of the proposed  algorithm.  Generally,  the  paper  is  well  written  and  organized.  Thus,  I  will  recommend  several  clarifications  before acceptance of this paper which are listed below:

  1. According to the topic of this paper, the authors need to supplement some related references to emphasize the main contribution and innovation of the paper as for the effect of sampling interval and data transmission for active safety system of advanced control technique such as Simultaneous Feasible Local Planning and Path-Following Control for Autonomous Driving,Robust vibration control for active suspension system of in-wheel-motor-driven electric vehicle via μ-synthesis methodology.

Response from the authors

Thank you for your suggestions. We extended our literature reviews with suggested papers.

2.The detailed test platform and its layout, the detailed process of the test should be presented.

Response from the authors

We revised the manuscript to include the detailed test platform, and processes as follows:

Line 452

The sensor data, including chassis, vision, and radar, were collected based on the test drives of CAVs conducted in Sejong city. We conducted eight Eight test drives are con-ducted from August to November 2020., The total length of test route is 15.64 km and the total driving time was 21 hours. One test drive was divided into two sessions: normal driving and abnormal driving. In the normal driving session, the CAV traveled one prede-fined route without causing any safety-related events. In contrast, in the abnormal driving session, the CAV traveled the pre-defined route six times, while generating safety-related events such as SD, left and right tilt, and rapid acceleration. The pre-defined route is shown in Figure 7, and a sample set of the collected data can be found at: https://github.com/benchoi93/CAVTestDriveData. The architecture of this study is developed based on the cloud platform, Microsoft Azure. and the communications are based on WAVE.

3.The expression should be improved. The English needs an improvement. The paper has some typos which need to be checked and corrected in the revision.

Response from the authors

We thoroughly revised the paper for better expressions.

  1. The entire flowchart and presentation of results should be improved, these results are not clear.

Response from the authors

We thoroughly revised the paper for better presentation of the results.

5.The use of proposed sampling interval and data transmission algorithm rather than other techniques should be discussed, execution time comparison in tests should be provided.

Response from the authors

In this study, the execution was performed whenever data was loaded. In other words, if the data is transmitted once a second, it is executed once a second. However, as the reviewer said, research on the execution time is also necessary, so the contents of future research in Conclusion were revised as follows.

Line 738

In the future, the characteristics of more diverse safety indicators will be analyzed by expanding the results of this study for various applications such as evaluation of collision risk and control of automated vehicle and for other execution strategies such as execution time and calculation of safety indicator at in-vehicle device [58,59,60]. Furthermore, although this study assumed the same sampling interval and data transmission interval, the follow-up paper will also analyze the optimal method for data transmission from connected and automated vehicles when different sampling intervals and transmission intervals depending on its urgency are applied.

Round 2

Reviewer 2 Report

The paper needs to be proofread.

Author Response

we carefully revised the paper through proofreading.